| Open Peer Review | Microbial Ecology | Methods and Protocols

# SS-VIME: a single-source virome-microbiome extraction protocol toward comprehensive soil community analysis

Abdonaser Poursalavati,[1,2] Isabelle Laforest-Lapointe,[2] Mamadou Lamine Fall[1]

**ABSTRACT** Integrated analysis of soil microbiomes and their associated viromes is critical for understanding ecosystem function, yet is hampered by the profound spatial heterogeneity of soil, which introduces significant bias when using separate extraction workflows and/or subsampling strategies to capture fungal, bacterial, and viral communities. Here, we present single-source extraction for unified soil virome-microbiome profiling (SS-VIME), a protocol that overcomes this limitation. Based on extended cellulose column chromatography, this method sequentially elutes distinct DNA and double-stranded RNA (dsRNA) fractions from a single soil lysate. We validated the protocol using sterilized soil co-spiked with a ZymoBIOMICS microbial community standard and a synthetic viral dsRNA fragment. Sequencing confirmed that the DNA fraction accurately recovered the theoretical bacterial (16S rRNA gene) and fungal (ITS) community profiles, while the dsRNA fraction demonstrated highly specific recovery of the target viral signature. The protocol was then successfully applied to characterize the complex native communities in environmental soil samples. The SS-VIME protocol provides a streamlined approach for isolating high-quality nucleic acids suitable for downstream applications. By using dsRNA as a proxy for viral activity and eliminating subsample bias, this method provides a robust, accessible, and unified platform to investigate virus-host dynamics *in situ*, paving the way for a more holistic understanding of the soil microbiome.

**IMPORTANCE** The study of soil microbes and their viruses, which are central to ecosystem health, is fundamentally limited by technical barriers. Separate extraction workflows for each group introduce sampling bias, obscuring the true ecological relationships within soil's spatially complex micro-environments. Our single-source virome-microbiome extraction (SS-VIME) protocol directly overcomes this by efficiently recovering both microbial DNA and viral double-stranded RNA (dsRNA) from one sample. This unified approach is not only cost-effective but, by using dsRNA as a signature of viral activity, captures a more accurate and representative profile of the soil active virome. SS-VIME provides the foundation for robustly investigating how viruses modulate soil health, carbon cycling, and agricultural productivity, moving the field from correlational studies toward a direct, integrated view of the soil ecosystem.

**KEYWORDS** soil virome, viromics, soil microbiome, dsRNA, cellulose column chromatography, unified protocol, single-source extraction, nucleic acid extraction

Soil ecosystems are critical to global biome functioning, hosting a staggering diversity of microorganisms including bacteria, archaea, fungi, and the viruses that infect them (1, 2). While microbial communities are known to drive essential processes like biogeochemical cycling and support plant health, their viral counterpart, the virome, remains a relatively neglected component of soil ecology (2, 3). Viruses are the most abundant biological entities on Earth and act as powerful agents of microbial

**Peer Reviewer** Gareth Trubl, Lawrence Livermore National Laboratory Physical and Life Sciences Directorate, Livermore, California, USA

Address correspondence to Mamadou Lamine Fall, mamadoulamine.fall@agr.gc.ca.

The authors declare no conflict of interest.

See the funding table on p. 10.

mortality, horizontal gene transfer, and metabolic reprogramming (4–6). A comprehensive understanding of soil ecology, therefore, requires integrated analytical approaches that can simultaneously characterize microbial and viral communities and unveil their complex interactions (7–9). However, achieving this unified view has been historically hindered by significant methodological barriers (10–12).

Current methods for studying soil microbial communities often rely on fragmented workflows targeting either microbial DNA for host profiling, viral particles for viromics, or total RNA for metatranscriptomics (13–16). This separation is profoundly problematic for soil, as decoupling the virome from the microbiome disrupts the spatial context needed to study virus-host interactions and quantify viral impacts on biogeochemical cycling (3, 17, 18). Indeed, such holistic approaches are essential to fully capture the complexity of multi-trophic interactions in the soil ecosystem (19). It is now well-established that spatial variability in soil is not random noise but rather a predictable, structured phenomenon, with a considerable portion of the variance in microbial populations and their geochemical activities occurring at the submillimeter scale (20, 21). In this context, where "millimeters matter," the practice of analyzing separate subsamples for different biological targets means that researchers are effectively studying different micro-communities (6, 22). This introduces significant sampling bias, fundamentally decoupling the viral data from its host microbial context to correlate their dynamics (2, 3, 23). Beyond this critical sampling issue, standard viromics approaches like virus-like particle (VLP) extraction are inherently biased toward encapsidated virions and often fail to capture intracellular viruses or those lacking traditional capsids, like many mycoviruses abundant in soil (18, 24, 25) (Fig. 1A). Furthermore, the low abundance of viral genetic material relative to the overwhelming amount of nucleic acids from other soil microbial members in metagenomic and metatranscriptomic data sets often obscures viral detection, necessitating separate workflows and resource-intensive analytical pipelines to characterize DNA and RNA viruses (12, 18, 26). A promising alternative lies in targeting double-stranded RNA (dsRNA), a stable molecular signature produced as a replicative intermediate or transcriptional byproduct by the vast majority of both RNA and DNA viruses (27–33), making it an excellent proxy for viral activity, an approach that has been successfully applied to characterize the virome in other complex matrices like plant tissues (25, 34–40).

Our research has previously established the foundational components for an integrated approach. First, we developed a robust and cost-effective method for extracting high-quality total nucleic acids from challenging soil matrices, overcoming common issues like humic acid contamination and the high cost of commercial kits (42). Specifically, this method mitigated humic acid co-extraction through the use of a CTAB-PVP extraction buffer combined with a phosphate buffer to optimize pH during lysis, strict temperature control to prevent humic acid carryover, and a PEG-NaCl precipitation step to selectively recover nucleic acids. This provided a high-quality starting lysate rich in microbial and viral nucleic acids. Building on this, we then optimized a cellulose column chromatography protocol specifically designed to isolate and purify viral dsRNA from these complex soil extracts (43). This second method proved highly effective for characterizing the soil dsRNA virome, revealing a diversity of RNA viruses often missed by other techniques (43). The effectiveness of this method was further underscored by its subsequent adaptation for characterizing the virome of other matrices like fungal mycelium and plant tissues (44–46). While these foundational protocols were successful, they were designed to isolate viruses and their host nucleic acids. Consequently, studying the associated host microbiome would still require a separate workflow from a different soil subsample, reintroducing the very issue of spatial bias that a unified method seeks to solve.

Here, we present a significant advancement that unifies these previous methods into a single, streamlined protocol: single-source extraction for unified soil virome-microbiome profiling (SS-VIME). This extended cellulose column chromatography protocol enables the sequential elution of distinct DNA and dsRNA fractions from a single soil

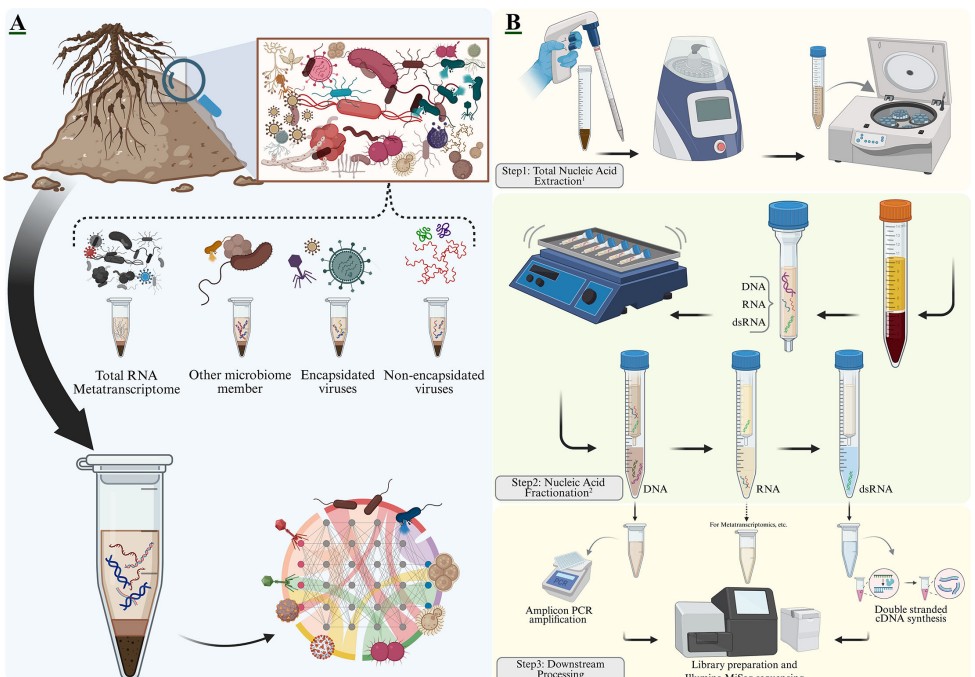

**FIG 1** Conceptual framework and workflow for single-source virome-microbiome profiling. (A) The conceptual basis and challenge addressed by the unified protocol. Soil is a spatially heterogeneous ecosystem containing a complex mixture of microbial members (e.g., bacteria and fungi) and a diverse virome, including both encapsidated and non-encapsidated viruses. Current approaches (dotted lines) are often fragmented, leading to inherent biases: separate extractions for microbial DNA and viral particles introduce spatial sampling bias, while total RNA extraction for metatranscriptomics captures all active members, but the low abundance of viral transcripts relative to host RNA often obscures viral detection. The SS-VIME method addresses these challenges by using a single soil lysate to ensure that both the microbial and viral communities are profiled from the same micro-environment, enabling robust network analysis of their correlated interactions. (B) The schematic workflow of the SS-VIME protocol, divided into three main stages. Step 1: Total Nucleic Acid Extraction[1]: A crude lysate containing the genetic material from the entire microbial community is generated from the soil sample through chemical and mechanical disruption. Step 2: Nucleic Acid Fractionation[2]: The lysate undergoes cellulose-column chromatography. The column differentially separates the nucleic acids, allowing for the sequential collection of a DNA fraction, an optional RNA fraction for functional studies (e.g., metatranscriptomics), and a dsRNA fraction, which serves as a proxy for the virome. Step 3: Downstream Processing: Each fraction is prepared for analysis. The DNA is used for amplicon PCR amplification to profile microbial communities using targeted amplicon sequencing (e.g., 16S rRNA gene/ITS). The dsRNA is converted to a sequencing library via double-stranded cDNA synthesis. The resulting libraries are then prepared for high-throughput sequencing (e.g., Illumina MiSeq). [1]Total nucleic acid extraction was performed according to Poursalavati et al. (41). [2]Cellulose-column chromatography was performed according to Poursalavati et al. (42). Schematic created with BioRender.com and edited in Inkscape. single-source virome-microbiome extraction (SS-VIME).

extract. The aims of this paper are to: (i) detail this unified protocol, which demonstrates substantial improvements in cost-effectiveness and efficiency when compared to existing techniques that require separate commercial kits for DNA and dsRNA extraction; (ii) demonstrate its capacity to recover high-quality DNA suitable for comprehensive bacterial (16S rRNA gene) and fungal (ITS) community profiling; and (iii) validate the efficient and specific capture of the viral dsRNA fraction using both environmental soil samples and spiked controls. For the first time, this method allows for the co-capture of microbial community DNA and a comprehensive viral signature from one sample, mitigating the sampling bias inherent in separate extractions and paving the way for more powerful and statistically robust investigations of correlated virus-host community dynamics in complex soil ecosystems.

## MATERIALS AND METHODS

### The unified extraction protocol

The SS-VIME protocol is a unified workflow designed to sequentially isolate DNA and dsRNA fractions from a single soil sample. The method's core principle relies on cellulose column chromatography, which differentially partitions nucleic acids based on ethanol concentration. Following a robust chemical and mechanical lysis step to release total nucleic acids, the lysate is conditioned with ethanol and applied to a cellulose column. Under these conditions, the DNA fraction passes through and is collected, while the dsRNA fraction remains bound to the cellulose matrix. The dsRNA is subsequently eluted using an ethanol-free buffer, resulting in two distinct nucleic acid pools from the same initial sample that are ready for downstream analysis.

The complete, step-by-step procedure, including all buffer recipes and critical notes, is publicly available at protocols.io (https://dx.doi.org/10.17504/protocols.io.kxygxp64zl8j/v1) and is also provided in File S1. The main stages of the protocol are illustrated in Fig. 1B. In brief, soil samples are homogenized in a PVP-amended CTAB buffer combined with phenol:chloroform:isoamyl alcohol (25:24:1) and β-mercaptoethanol to lyse microbial cells using a bead-beating homogenizer. Following centrifugation, the clarified supernatant is adjusted to ~16.2% ethanol and loaded onto a pre-equilibrated cellulose column, allowing for the collection of the DNA-containing flow-through. After a column wash to remove inhibitors and single-stranded RNA, the bound dsRNA is eluted with pure water. The eluted dsRNA fraction is then treated with DNase I and RNase T1 to remove residual DNA and single-stranded RNA. Finally, both the DNA and treated dsRNA fractions are separately concentrated via PEG-NaCl precipitation, washed with 75% ethanol, and resuspended, yielding purified nucleic acids suitable for downstream applications such as PCR and high-throughput sequencing.

### Validation experiments

#### Environmental soil samples

A total of eight environmental soil samples were collected from a vineyard on mineral soil at an Agriculture and Agri-Food Canada (AAFC) experimental farm in Frelighsburg, Québec. The soil is classified as the Blandford series (loam texture), characterized by a mean pH of 5.82, an organic matter content of 7.7%, and a C:N ratio of 10.5. Detailed physicochemical properties for both zones are provided in Table S1. To capture spatial micro-heterogeneity, sampling was performed at depths of 5–15 cm, targeting both the rhizosphere and the adjacent bulk soil zones. Immediately following collection, samples were placed in sterile 50 mL Falcon tubes without the addition of preservation buffers, transported on ice to the laboratory, and stored frozen at −20°C until extraction (~6 months). These samples served as the primary material for demonstrating the protocol's efficacy in co-extracting DNA and dsRNA from a complex environmental matrix.

#### Spike-in controls

Two distinct control soil matrices were created: one by pooling the environmental bulk soil samples (Control 1, C1) and another by pooling the rhizosphere samples (Control 2, C2). These matrices were then sterilized by drying at 60°C for 24 h, followed by three autoclave cycles, with 24-h incubation intervals. This tyndallization approach maximizes the elimination of heat-resistant spores and fragments background DNA (47, 48). To verify the efficacy of the sterilization, a negative control extraction was performed on a non-spiked aliquot of the pooled sterilized matrix (C1/C2 mix), followed by PCR amplification using the 16S rRNA gene and ITS primers. Each sterilized matrix was then co-spiked with two standards. First, 75 µL of the ZymoBIOMICS Microbial Community Standard (D6300; Zymo Research, Irvine, CA, USA) was added to assess DNA recovery and profiling accuracy. This cellular standard mimics a mixed community comprising three gram-negative and five "tough-to-lyse" gram-positive bacteria (*Listeria monocytogenes*,

*Pseudomonas aeruginosa*, *Bacillus subtilis*, *Escherichia coli*, *Salmonella enterica*, *Lactobacillus fermentum*, *Enterococcus faecalis*, and *Staphylococcus aureus*) and two "tough-to-lyse" fungal species (*Saccharomyces cerevisiae* and *Cryptococcus neoformans*), representing the phyla Firmicutes, Proteobacteria, Ascomycota, and Basidiomycota. Second, 50 µL of a synthetic ~500 bp dsRNA fragment (~11 ng/µL) from the tomato brown rugose fruit virus (ToBRFV), synthesized by Gene Universal (Newark, DE, USA), to validate dsRNA capture. These spiked controls were processed through the entire wet-lab and bioinformatic workflow. Additionally, to serve as a benchmark for the downstream analysis, a portion of the unspiked dsRNA standard (Raw Input) was also processed in parallel, starting from the cDNA synthesis stage.

## Downstream analysis

### *Library preparation and sequencing*

Following extraction, the yield and purity of the DNA and dsRNA fractions were assessed. Quantification was performed using a Qubit 4.0 Fluorometer (Invitrogen, Waltham, MA, USA) with the Qubit dsDNA High Sensitivity (HS) Assay Kit, utilizing 1 µL of sample for both DNA and dsRNA fractions. Purity ratios (A260/280 and A260/230) were determined via spectrophotometry (Nanodrop 2000; Thermo Fisher Scientific, Waltham, MA, USA) using 2 µL of sample. The successful recovery of the spiked standards from the control samples was visually confirmed by agarose gel electrophoresis. The dsRNA fraction was converted into double-stranded cDNA using our optimized synthesis protocol, which is based on random priming with SuperScript IV Reverse Transcriptase (Thermo Fisher Scientific). This complete method is detailed in File S2 and on protocols.io ( https://dx.doi.org/10.17504/protocols.io.5qpvobdybl4o/v1). Sequencing libraries were then constructed from the resulting cDNA using the Nextera DNA Library Preparation Kit (Illumina, San Diego, CA, USA). For the DNA fraction, amplicon libraries were generated by targeting the bacterial 16S rRNA gene V4–V5 region (primers 515F-Y/926R) and the fungal Internal Transcribed Spacer 1 (ITS1) region (primers ITS5/ITS2) in a one-step PCR with a custom dual-indexed adapter approach (49) (Table S2). All resulting microbial amplicon and viral cDNA libraries were sequenced at the AAFC Research and Development Centre (Saint-Jean-sur-Richelieu, QC, Canada) on an Illumina MiSeq platform (v3 reagent kits; Illumina) to generate $2 \times 300$ bp paired-end reads.

### *Bioinformatic analysis*

The resulting sequencing data were processed with distinct pipelines. Raw paired-end reads from the amplicon libraries were first demultiplexed using Cutadapt (50) (v4.8; parameters: -e 0.15 --no-indels) and subsequently processed with the nf-core/ampliseq pipeline (51) (v2.13.0). This workflow performs quality control, merges reads, and utilizes DADA2 for amplicon sequence variant (ASV) calling (truncation parameters: 16S: 230/160 bp; ITS: 245/200 bp), followed by taxonomic classification and diversity analysis within a QIIME2 framework.

The dsRNA-derived reads were analyzed using the SOVAP workflow (41, 43). This workflow includes: quality trimming with Fastp (52) (v0.23.2; parameters: window size 4, mean quality 15), decontaminated against non-viral sequences using Centrifuge (53) (v1.0.4; parameter: --min-hitlen 50), and assembled into contigs with MEGAHIT (54) (v1.2.9; parameters: --presets meta-large, --min-contig-len 500). Viral contigs were then identified using geNomad (55) (v1.11.1; parameters: end-to-end, --min-score 0.7). To validate the robustness of this identification strategy, a cross-comparison was performed against a suite of viral identification (VirSorter2, DeepVirFinder, ViraLM, and TransGINmer), quality assessment (CheckV), and classification tools (vConTACT3) (56–61). All tools were executed using default parameters. These comparative results and a consensus matrix are detailed in File S3. Following identification, viral contigs were clustered into viral operational taxonomic units (vOTUs) using CD-HIT (62) (v4.8.1; parameters: -c 0.95 -aS 0.85, corresponding to 95% identity and 85% alignment

coverage). Taxonomic annotation was performed by searching against the IMG/VR v4 and NCBI RefSeq viral databases with DIAMOND (63) (v2.1.4; blastx default parameters). Finally, the relative abundance of each vOTU was determined by mapping reads back to the viral contigs using BWA-MEM (64) (v0.7.17; default parameters).

## RESULTS

### Single-source extraction successfully partitions DNA and dsRNA fractions

The SS-VIME protocol successfully partitioned and recovered DNA and dsRNA fractions from a single soil lysate. The method yielded high-purity DNA, exhibiting mean A260/280 ratios of 1.88 and A260/230 ratios of 1.95. Normalized yields varied by soil zone, with bulk soil samples yielding 8.36 ± 1.58 µg/g and rhizosphere samples yielding 15.73 ± 2.37 µg/g. The corresponding dsRNA fraction yielded 10.08 ± 4.69 ng/g in bulk soil and 24.56 ± 6.56 ng/g in the rhizosphere, consistent with the expected yield of dsRNA (34). The protocol's efficacy was validated using sterilized soil controls (C1 and C2) spiked with known standards (Fig. 2). The recovered DNA fraction from the Zymo standard appeared as a high-molecular-weight smear, characteristic of intact genomic DNA (Fig. 2A). Similarly, the dsRNA fraction, eluted from the cellulose column, yielded a clean, distinct band corresponding to the ~500 bp synthetic viral dsRNA fragment (Fig. 2B). The quality of these fractions was confirmed by their suitability as templates in downstream enzymatic reactions. Robust amplicons for both the 16S rRNA gene and ITS regions were generated from the DNA fraction (Fig. 2C), while the dsRNA was successfully reverse-transcribed into double-stranded cDNA using a random primer-based method, visualized as a smear (Fig. 2D). As a final confirmation step prior to sequencing, the identity and suitability of this cDNA was verified by amplifying the target fragment using specific primers (Table S3), which yielded a clean amplicon of the correct size (Fig. 2E). The effectiveness of the sterilization process was confirmed by the absence of amplifiable DNA in non-spiked control extracts (Fig. S1).

### The DNA fraction accurately recovers microbial community profiles

To validate the quantitative accuracy of the DNA fraction, the spiked mock community controls were analyzed via targeted amplicon sequencing. The 16S rRNA gene profiling successfully identified all eight bacterial members of the Zymo standard in both the bulk (Mock_C1) and rhizosphere (Mock_C2) controls. The observed relative abundances showed high fidelity to the theoretical composition (Fig. 3A), with only minor deviations consistent with known PCR biases, such as a slight underrepresentation of *Staphylococcus* and overrepresentation of *Pseudomonas*. This demonstrates that the protocol introduces minimal extraction or amplification bias for key bacterial taxa. Similarly, ITS sequencing accurately recovered the two fungal species, *S. cerevisiae* and *C. neoformans*, in proportions consistent with the theoretical standard, confirming the method's reliability for fungal profiling (Fig. 3B). Application of the SS-VIME protocol to the vineyard soil samples successfully characterized the complex native microbial communities. Bacterial community profiles were dominated by the phyla Acidobacteriota, Proteobacteria, and Actinobacteriota across all samples (Fig. 3C). Fungal communities were primarily composed of members of Ascomycota and Mortierellomycota (Fig. 3D). The complete taxonomic profiles and relative abundances for all samples are provided in File S4.

### The dsRNA fraction specifically recovers the virome signature

The specificity and efficacy of the dsRNA fraction for virome recovery were validated through the sequencing of the spiked controls. The taxonomic profile of the dsRNA recovered from the spiked controls (Spiked_C1 and Spiked_C2) precisely matched that of the directly sequenced synthetic dsRNA standard (Raw_Input). At the species level, 100% of reads were classified within the genus *Tobamovirus*, mainly to tomato brown rugose fruit virus (TBRFV) with a small proportion assigned to tobacco mosaic virus (Fig. 4A and

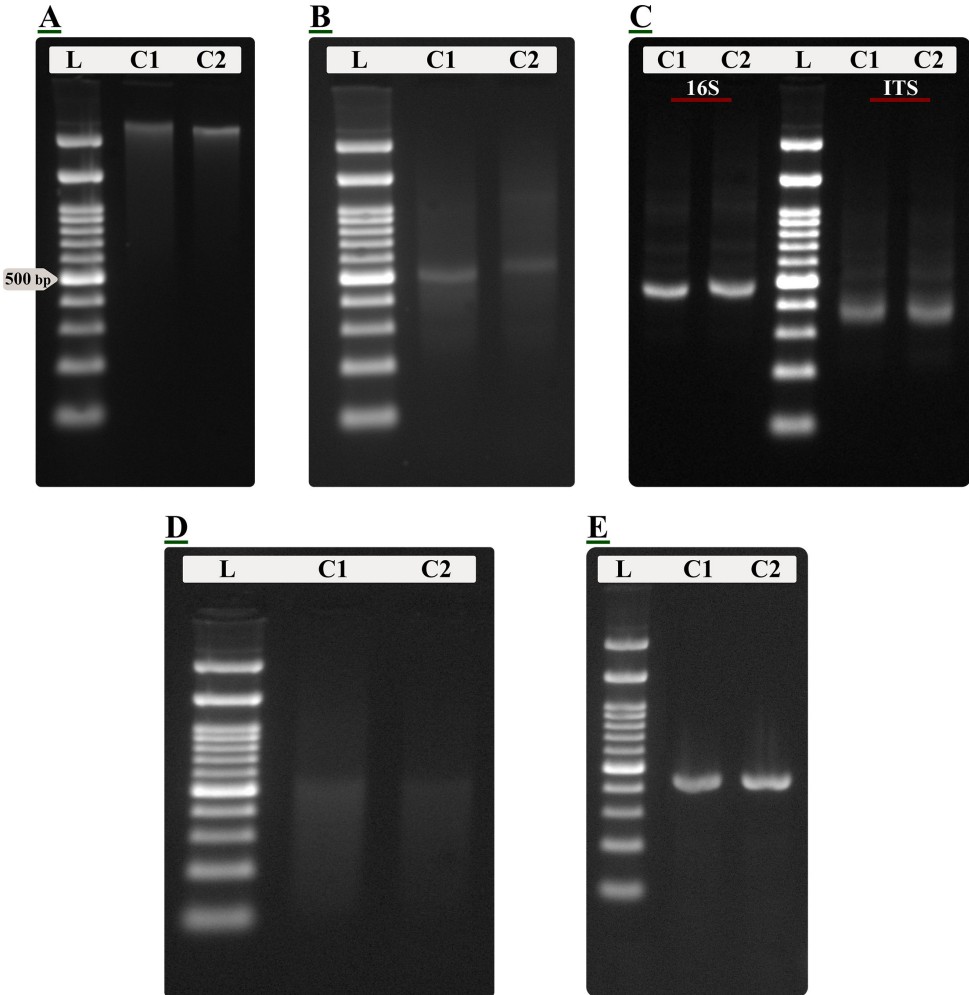

**FIG 2** Validation of the SS-VIME protocol using spiked control samples. Agarose gel electrophoresis of nucleic acids extracted and processed from two sterile soil controls (C1: mixed bulk soil samples; C2: mixed rhizosphere soil samples) spiked with a Zymo microbial standard and a synthetic viral dsRNA fragment. (A) High molecular weight DNA recovered in the DNA fraction. (B) The purified dsRNA fraction eluted from the column, showing a distinct band at the expected size (~500 bp). (C) Successful PCR amplification of 16S rRNA gene and ITS markers from the DNA fraction shown in panel (A). (D) Double-stranded cDNA synthesized from the dsRNA template shown in panel (B). (E) Confirmatory PCR product amplified from the cDNA (panel D) using primers specific to the synthetic dsRNA target. L, 100 bp DNA ladder; the 500 bp band is indicated for reference. All gels are 1% agarose in SB buffer. single-source virome-microbiome extraction (SS-VIME).

B). This result confirms the protocol's high fidelity and specificity in capturing the target dsRNA molecule from a complex soil matrix with minimal background noise.

When applied to environmental samples, the method revealed a complex and diverse native soil virome. At the class level, communities were composed of numerous distinct viral taxa, ranging from the DNA-based tailed bacteriophages (*Caudoviricetes*) to a variety of RNA viral classes, including *Duplopiviricetes*, *Resentoviricetes*, *Pisoniviricetes*, and *Leviviricetes* (Fig. 4C). Analysis at the realm level showed that the recovered virome was composed of viruses from *Duplodnaviria*, *Riboviria*, and *Monodnaviria* (Fig. 4D). A detailed taxonomic breakdown of the environmental virome is available in File S5.

## DISCUSSION

In this study, we developed and validated SS-VIME, a single-source extraction protocol that successfully partitions microbial DNA and viral dsRNA from a single soil sample. This unified approach offers a strategy to mitigate the potential sampling bias inherent to

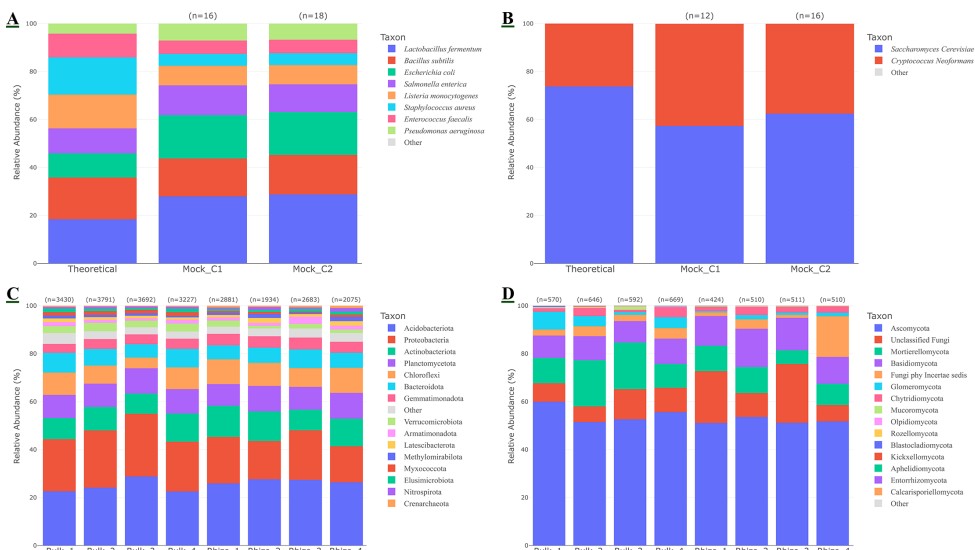

**FIG 3** Validation of microbial community profiling using the DNA fraction from spiked controls and environmental soil samples. Stacked bar charts show the relative abundance of microbial taxa recovered. (A) Bacterial community composition (species level) recovered from two mock controls (Mock_C1 and Mock_C2, representing spiked sterile bulk and rhizosphere soil, respectively) compared to the theoretical composition of the Zymo standard. (B) Fungal community composition (species level) recovered from the same mock controls. (C) Relative abundance of the top 15 most abundant bacterial phyla across the four bulk and four rhizosphere environmental soil samples. (D) Relative abundance of the top 15 most abundant fungal phyla across the environmental soil samples. The total number of observed amplicon sequence variants (ASVs) for each sample is indicated above the respective bar.

spatially heterogeneous soil ecosystems, where "millimeters matter" (6, 65). In such contexts, the use of separate subsamples for different targets can confound the analysis of virus-host community dynamics. Our validation with spiked controls provides robust proof-of-concept: the DNA fraction yielded a microbial profile that accurately mirrored the theoretical composition of the Zymo standard (Fig. 3A and B), while the dsRNA fraction demonstrated the highly specific recovery of the synthetic viral fragment tomato brown rugose fruit virus (Fig. 4A and B), with low hits assigned tobacco mosaic virus being an expected result due to the high sequence similarity between it and the synthetic fragment of TBRFV.

Furthermore, its application to vineyard soils yielded microbial profiles dominated by phyla such as Acidobacteriota, Proteobacteria, and Ascomycota (Fig. 3C and D), consistent with findings from similar agricultural and vineyard soil (66–69). Crucially, the dsRNA fraction captured a broad virome spanning the realms *Duplodnaviria* (DNA viruses), *Riboviria* (RNA viruses), and *Monodnaviria* (ssDNA viruses) (Fig. 4D). The ability to recover dsRNA signatures from RNA and DNA viruses highlights the potential of this method to serve as a proxy for the active virome, herein defined as the subset of viruses actively undergoing transcription and replication (27, 29, 32, 33, 70–73). While the spike-in confirms the chemical efficiency of dsRNA capture, the method's capacity to recover a phylogenetically diverse virome relies on effective host lysis (here validated by the Zymo standard) and was demonstrated by the broad viral taxa recovered from the environmental samples.

The SS-VIME protocol also offers versatility for broader multi-omics investigations. The high-quality DNA fraction allows for the profiling of diverse microbial guilds beyond bacteria and fungi (e.g., arbuscular mycorrhizal fungi, protists, or archaea), while the optional total RNA fraction recovered during the wash step suggests the potential for future integration with metatranscriptomic workflows to link community composition and viral replication to host gene expression.

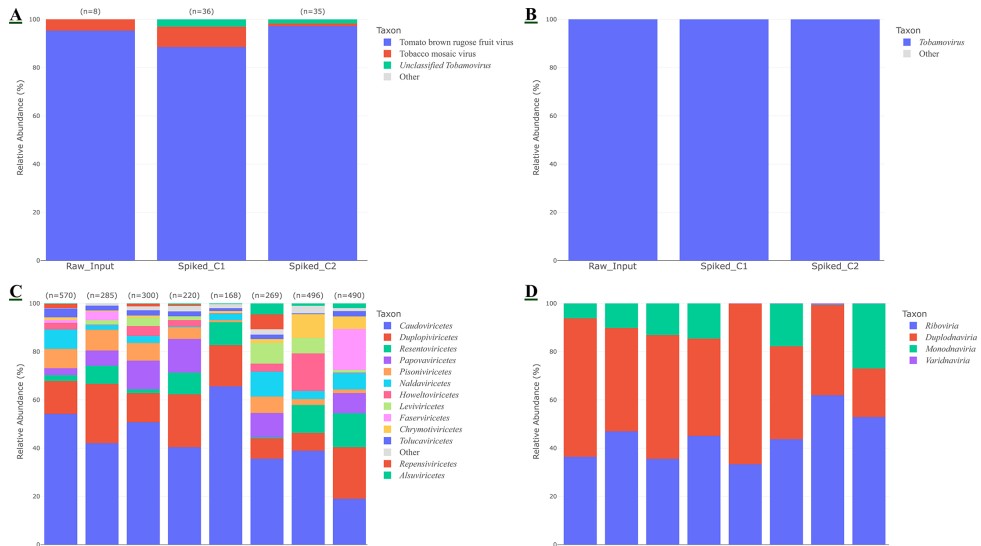

**FIG 4** Validation and application of virome profiling using the dsRNA fraction. Stacked bar charts show the relative abundance of viral taxa recovered from spiked controls and environmental soil samples. (A, B) Taxonomic composition of the dsRNA fraction recovered from two spiked sterile soil controls (Spiked_C1 and Spiked_C2) compared to the directly sequenced synthetic dsRNA standard (Raw_Input). Profiles are shown at the (A) species and (B) genus levels, demonstrating the specific recovery of the spiked *Tobamovirus*. (C, D) viral community composition in the eight environmental soil samples. To highlight known viral diversity, unclassified viruses were excluded from these plots. Profiles are shown at the (C) class level (displaying taxa with >2% relative abundance) and (D) realm level. The total number of identified viral operational taxonomic units (vOTUs) for each sample is indicated above the bars in panels (A and C). The complete taxonomic profiles, including unclassified taxa, are available in File S5.

While this protocol represents a significant technical advance, it is important to acknowledge its scope. The method is optimized to capture dsRNA as a broad and stable proxy for the viral activity. Consequently, the dsRNA fraction is not designed to detect latent prophages or the inert virions of DNA and ssRNA viruses, though it will capture the genomic dsRNA from intact dsRNA virions. Regarding DNA viruses, detection relies on the capture of dsRNA intermediates generated via convergent bidirectional transcription, transcriptional readthrough, or host-mediated synthesis (74–78); consequently, these viruses may be underrepresented depending on their transcriptional state. Future work should focus on testing the protocol's performance across a wider variety of challenging matrices, particularly mineral-rich soils with high iron or clay content, to fully define its operational boundaries. Additionally, while the current validation utilized amplicon profiling for community analysis, applying shotgun metagenomics to the DNA fraction represents a critical next step. This would allow for the detection of lysogenic prophages and enable more precise genomic linking between specific viral lineages and their microbial hosts. Furthermore, while our synthetic spike-in validated the capture chemistry, future benchmarking with mock communities of diverse viruses would further refine the resolution of recovery rates across different viral families. Nonetheless, by providing a robust and accessible method for integrated virome-microbiome analysis, SS-VIME lowers the barrier to entry for conducting more holistic studies of the soil ecosystem, moving the field beyond correlational observations from disparate data sets toward a more integrated analysis of virus-host co-occurrence patterns.

## Conclusion

We present the SS-VIME protocol, a novel and robust method for the single-source extraction of nucleic acids for microbial and viral community profiling from soil. By enabling the co-extraction of high-quality microbial DNA suitable for microbial marker

gene profiling and a comprehensive viral dsRNA signature from a single source, this protocol mitigates the fundamental challenges of sampling bias caused by spatial heterogeneity. Notably, the method successfully captured diverse mycoviruses, including members of the *Polymycoviridae* and *Betaormycoviridae* families, validating its potential for elucidating fungal-viral interactions and recovering distinct viral lineages within the soil matrix. Ultimately, SS-VIME provides an accessible and unified platform for investigating the intricate ecological correlations between viruses and their hosts, paving the way for a more integrated and holistic understanding of the soil microbiome's structure, function, and dynamics.

## ACKNOWLEDGMENTS

We thank the staff at the Agriculture and Agri-Food Canada Saint-Jean-sur-Richelieu Research and Development Centre (AAFC-SJR-RDC) for their support throughout this project. We are particularly grateful for the direct technical contributions of Dong Xu and Pierre Lemoyne, for their expertise in library preparation and sequencing; and Benjamin Mimee, Jacynthe Masse, Pierre-Yves Véronneau, and Guillaume Trépanier for their valuable assistance with the amplicon sequencing methodology. We also extend our thanks for field sampling assistance to Pierre Lemoyne, Vahid Jalali Javaran, Sarah Drury, and Amadou Sidibe.

This project was critically supported by the management of the experimental farms by the teams of Eric Courchesne and Martin Robidoux, and by the essential support from the teams of Vicky Toussaint, Mélanie Maheu, and Dominique Roussel.

Conceptualization and experimental design were performed by A.P. and M.L.F. A.P. conducted the laboratory experiments, performed the bioinformatic analysis, and created the data visualizations. The original manuscript was drafted by A.P. and reviewed by M.L.F. I.L.-L. and M.L.F. provided supervision and critically reviewed and edited the manuscript. All authors contributed to the article and approved the submitted version.

## AUTHOR AFFILIATIONS

[1]Saint-Jean-sur-Richelieu Research and Development Centre, Agriculture and Agri-Food Canada, Saint-Jean-sur-Richelieu, Quebec, Canada
[2]Centre SÈVE, Department of Biology, Université de Sherbrooke, Sherbrooke, Quebec, Canada

## AUTHOR ORCIDs

Abdonaser Poursalavati  http://orcid.org/0000-0002-6238-6012
Mamadou Lamine Fall  http://orcid.org/0000-0002-1747-6481

## FUNDING

| Funder | Grant(s) | Author(s) |
| --- | --- | --- |
| Agriculture and Agri-Food Canada | J-003454, J-003174 | Abdonaser Poursalavati |
| | | Mamadou Lamine Fall |

## AUTHOR CONTRIBUTIONS

Abdonaser Poursalavati, Conceptualization, Data curation, Formal analysis, Methodology, Validation, Visualization, Writing – original draft, Writing – review and editing | Isabelle Laforest-Lapointe, Supervision, Writing – review and editing | Mamadou Lamine Fall, Conceptualization, Data curation, Funding acquisition, Investigation, Methodology, Project administration, Resources, Software, Supervision, Validation, Writing – review and editing

## DATA AVAILABILITY

The raw 16S rRNA gene, ITS, and shotgun metaviromic sequencing data generated for this study have been deposited in the NCBI Sequence Read Archive (SRA) under the umbrella BioProject accession PRJNA1305474. A detailed list of individual SRA and BioSample accession numbers, along with their corresponding sample IDs and data types, is provided in Table S4.

## ADDITIONAL FILES

The following material is available online.

### Supplemental Material

**File S1 (Spectrum03323-25-s0001.pdf).** SS-VIME: Single-Source Virome-Microbiome Detailed Extraction Protocol for simultaneous profiling of viral, bacterial, and fungal communities from a single soil sample.
**File S2 (Spectrum03323-25-s0002.xlsx).** This spreadsheet is a reagent calculator for ds-cDNA synthesis protocol. This method is the official downstream application for the dsRNA fraction isolated using our SS-VIME (Single-Source Virome-Microbiome Extraction) workflow.
**File S3 (Spectrum03323-25-s0003.xlsx).** Statistics summary of pipelines performances.
**File S4 (Spectrum03323-25-s0004.xlsx).** Microbial profiling outputs.
**File S5 (Spectrum03323-25-s0005.xlsx).** Virome profile and abundance.
**Figure S1 (Spectrum03323-25-s0006.docx).** Negative Control PCR. Agarose gel electrophoresis of PCR products from the negative control extraction. The input sample was a non-spiked, sterilized mixture of pooled bulk and rhizosphere soil (C1/C2). Lanes verify the absence of amplifiable DNA using 16S rRNA and ITS primers, compared to a no-template control (NTC). L: 100 bp DNA ladder.
**Tables S1 to S4 (Spectrum03323-25-s0007.xlsx).** Table S1: Physicochemical properties of the Frelighsburg vineyard soil samples.
Table S2: Primers used for 16S rRNA and ITS amplicon sequencing.
Table S3: Primers used for the PCR-based validation of the synthetic ToBRFV dsRNA spike-in.
Table S4: Summary of sequencing data and NCBI accession numbers for all samples.

### Open Peer Review

**PEER REVIEW HISTORY (review-history.pdf).** An accounting of the reviewer comments and feedback.

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
