## [Reviewer comments · Microbiology Spectrum]

Microbiology Spectrum

SS-VIME: A Single-Source Virome-Microbiome Extraction Protocol Toward Comprehensive Soil Community Analysis

Abdonaser Poursalavati, Isabelle Laforest-Lapointe, and Mamadou Lamine Fall

Corresponding Author(s): Mamadou Lamine Fall, Agriculture and AgriFood Canada

Review Timeline:

Submission Date:	October 15, 2025
Editorial Decision:	December 8, 2025
Revision Received:	January 15, 2026
Accepted:	January 26, 2026

Editor: Blaire Steven

Reviewer(s): Disclosure of reviewer identity is with reference to reviewer comments included in decision letter(s). The following individuals involved in review of your submission have agreed to reveal their identity: Gareth Trubl (Reviewer #1)

Transaction Report:

DOI: <https://doi.org/10.1128/spectrum.03323-25>

Re: Spectrum03323-25 (**SS-VIME: A Single-Source Virome-Microbiome Extraction Protocol Toward Comprehensive Soil Community Analysis**)

Dear Dr. Mamadou Lamine Fall:

Thank you for the privilege of reviewing your work. Below you will find my comments, instructions from the Spectrum editorial office, and the reviewer comments.

Revision Guidelines

Sincerely,
Blair Steven
Editor
Microbiology Spectrum

Reviewer #1 (Comments for the Author):

This manuscript presents and validates a new technical protocol, SS-VIME, designed to simultaneously extract microbial DNA and viral double-stranded RNA from a single soil sample. The core goal of SS-VIME is to address the fundamental challenge of spatial heterogeneity and associated sampling bias when studying virus-host dynamics in the soil's complex micro-environment. The authors successfully demonstrated proof-of-concept using spiked controls to show accurate microbial community (via amplicons) and highly specific viral signal (dsRNA fraction) recovery. Application to a vineyard soil confirms the method's ability to yield distinct microbial and virome profiles. I really appreciate the lengths the authors have gone through to make the methods

easy to follow and available to others.

The methodology used in this paper does not offer the necessary empirical support for many of the strong ecological claims made in the manuscript, since many of the claims are about heterogeneity and complexity, which the authors remove when sampling with marker genes. There was no control (spike-ins were not controls here but rather proof-of-concept) or comparison to other methodology to show improvement. The paper compares their methodology to multi-omics analyses and viromics when these methods are not used or compared to. The authors should work to revise the manuscript to show a new way to extract viruses from soils along with amplicons.

Major issues

1. The central claim is that the protocol "directly addresses the critical challenge of spatial heterogeneity." However, spatial heterogeneity was not directly assessed in this study, but rather, the authors demonstrated accurate recovery of spiked nucleic acids and recovery of viral genomes. To make this claim, a comparison is needed showing reduced variance (or improved correlation between virus/host pairs) in SS-VIME data versus data generated from split/separate subsamples. I recommend revising the text to state that the protocol works towards reducing spatial heterogeneity and sampling bias but has not definitively removed or overcome it.
2. The authors repeatedly use dsRNA as a "comprehensive proxy for the active virome." While I agree dsRNA is a good marker, the claim of capturing the "active virome" is difficult to support without functional validation. Replication takes place within a microbial cell and not all processes captured by dsRNA lead to the release of new virions or significant ecosystem impact. Increased transcriptional reads in metatranscriptomes versus controls are typically required to conclusively show metabolic activity. I recommend tempering this claim throughout the text.
3. The spike-in validation is limited. No ssDNA or dsDNA viruses were spiked in, and the RNA spiked-in was only dsRNA. This does not fully mimic the stability or extraction efficiency challenges posed by encapsidated ssRNA, mRNA, or DNA viruses.

Minor issues

1. The assessment of microbial community structure based on amplicon data has inherent limitations. While useful for quality control, marker genes cannot be used to link a virus to a host. A large part of the described bias and issues in the introduction of this paper comes from using metagenomics and metatranscriptomics to capture microbial and viral genomes to use genomic features to link them and omics is not used here at all. The amplicon work does allow for a great assessment of contamination of the virome data and is a cheap, quick means of initial characterization.
2. The Methods lack important descriptive details about the soil texture and chemistry of the vineyard soil.
3. I am not sure what "sterilization" of the soil added to this work. If this has been validated for soil, a citation that demonstrates its effectiveness should be provided. As a reviewer, I always argue that soils cannot be truly sterilized and attempts fundamentally alter the soil matrix.

Line-specific comments

Keywords "Soil Virome, Viromics, Soil Microbiome, dsRNA, Cellulose Column Chromatography, Unified Protocol, Single-Source Extraction"

I would remove viromics, because this refers to specific methodology (e.g., virus particle separation) that was not done here. I would add Nucleic Acid Extraction

Introduction

Lines 58-59 "This separation is profoundly problematic for soil (Geisen 2021, Liang, Radosevich et al. 2024, Hazard, Anantharaman et al. 2025)."

Please add some more text to this sentence that summarizes what these citations include.

Figure 1 legend "Downstream Processing: Each fraction is prepared for analysis. The DNA is used for amplicon PCR amplification to profile microbial communities using targeted amplicon sequencing (e.g., 16S/ITS)."

For every instance where 16S is used, please always have "16S rRNA gene" or "16S rRNA amplicon".

Methods

Lines 164-167 "A total of eight environmental soil samples were collected from a vineyard on mineral soil at an Agriculture and Agri-Food Canada (AAFC) experimental farm in Frelighsburg, Québec. To capture spatial micro-heterogeneity, sampling was performed at depths of 5-15 cm, targeting both the rhizosphere and the adjacent bulk soil zones."

Where is the description of the soil (e.g., texture,...)? This is where the complexity and chemistry comes in and it is absent from this paper.

Lines 167-169 "Immediately following collection, all samples were transported on ice to the laboratory and stored frozen at -20 {degree sign}C to preserve microbial community structure and ensure nucleic acid integrity until extraction could be performed." Please describe the container the soils was stored in, if any buffers were added, and how long the soil was at -20 before nucleic

acid extraction.

Lines 174-177 "These matrices were then sterilized by drying at 60{degree sign}C followed by three autoclave cycles, with 24-hour incubation intervals between cycles to allow for the germination and subsequent elimination of heat-resistant endospores." If this has been validated, then please provide a citation that demonstrates effectiveness. I always argue that soils cannot be sterilized and attempts to sterilize it just alter the microbiome and virome.

Lines 177-179 "Each sterilized matrix was then co-spiked with two standards: 75 µL of the ZymoBIOMICS Microbial Community Standard (D6300; Zymo Research, Irvine, CA, USA) to assess DNA recovery and community profile accuracy". Please briefly list the taxa in the community. It would be great to know if it is only bacterial and the phyla that is represented. Additionally, what buffer are the spike-ins in? This was also added to the soil and would stabilize the spike-in and not represent true soil conditions.

Lines 187-190 "Following extraction, the yield and purity of the DNA and dsRNA fractions were assessed using fluorometry (Qubit; Invitrogen, Waltham, MA, USA) for accurate quantification and spectrophotometry (Nanodrop 2000; Thermo Fisher Scientific, Waltham, MA, USA) for purity ratios."

Please list what version of Qubit, standards used, the amount of nucleic acid used per run, and any changes for DNA/RNA measurements. For the Nanodrop please list the purity ratios that were assessed.

Lines 198-200 "For the DNA fraction, amplicon libraries were generated by targeting the bacterial 16S rRNA V4-V5 region (primers 515F-Y/926R)"

Please add "gene" to 16S rRNA.

Line 207 "Cutadapt".

For every tool, please list the version and whether default parameters were used.

Lines 214-216 "Viral contigs were then identified using geNomad (Camargo, Roux et al. 2024) and clustered into viral operational taxonomic units (vOTUs) at 95% identity using with CD-HIT (Fu, Niu et al. 2012)."

Please list what the alignment fraction is for clustering. If it is 100%, then state that.

Results

Lines 224-225 "The method yielded high-purity DNA, with concentrations ranging from 200 to 600 ng/µl and A260/280 ratios of approximately 1.8-1.9."

When extracting from soil, the concentrations need to be normalized and presented in weight of soil, typically per gram. When presenting a range please provide the standard deviation of a typical DNA extraction. Was the A230:260 not measured? Please provide if possible as this ratio also informs us on how clean the DNA is and how reliable the concentration is.

Discussion

Lines 302-306 "In this study, we developed and validated SS-VIME, a single-source extraction protocol that successfully partitions microbial DNA and viral dsRNA from a single soil sample. This unified approach directly addresses the critical challenge of spatial heterogeneity in soil ecosystems, where 'millimeters matter' (Bi, Yu et al. 2021, Roux and Emerson 2022), and the use of separate subsamples for different targets can obscure true ecological relationships."

I would revise this text. Spatial heterogeneity was not assessed here, but rather spiking in nucleic acid and recovering it. Metagenomics and a comparison to another method is needed to state this claim.

Lines 316-320 "Crucially, the dsRNA fraction captured a broad virome spanning the realms Duplodnaviria (DNA viruses), Riboviria (RNA viruses), and Monodnaviria (ssDNA viruses) (Fig. 4D). The ability to recover dsRNA signatures from RNA and DNA viruses underlines the power of this method as a comprehensive proxy for the active virome, which includes viruses actively undergoing transcription and replication"

While I agree this is another method to capture soil viruses, it has not been compared to other studies to show improvement. Additionally, I do not agree that the active virome is captured, even with specifying active denotes transcription and replication. Replication takes place within a microbial cell which does not always result in cell lysis and release of viruses. Additionally, it is well known that DNA and RNA can be stable in soils and thus increased transcriptional reads in metatranscriptomes vs. controls are needed to show activity.

Lines 323-331 "The SS-VIME protocol also offers significant versatility for broader multi-omics investigations. The optional total RNA fraction, recovered during the column wash step, provides a valuable template for metatranscriptomic analysis. This creates a powerful opportunity to link community composition (DNA) and viral replication (dsRNA) directly to host gene expression (RNA) within a single experimental framework. Moreover, the high-quality DNA fraction is not limited to bacterial and fungal profiling; it serves as a comprehensive resource for characterizing other crucial microbial guilds. With the appropriate primers, this DNA can be used to profile communities of arbuscular mycorrhizal fungi (AMF), protists, or archaea, making this protocol a flexible platform for a wide range of ecological questions."

I would reduce this to a single sentence that states this method is applicable to multi-omics approaches. This work was not tested here and is a proposed use.

Lines 332-335 "While this protocol represents a significant technical advance, it is important to acknowledge its scope. The method is optimized to capture dsRNA as a broad and stable proxy for the active virome. Consequently, it is not designed to detect latent prophages or the inert virions of DNA and ssRNA viruses, though it will capture the genomic dsRNA from intact dsRNA virions."

I would also add text about ssDNA viruses due to library methods not capturing these, viruses actively undergoing lytic infection, and not capturing microbial genomes.

Lines 335-338 "Future work should focus on testing the protocol's performance across a wider variety of challenging soil types, such as those with extremely high clay or organic matter content, and sequencing the RNA fraction to fully define its operational boundaries."

High mineral soil, especially ones with lot of iron complexes create major issues, especially for RNA.

Conclusion

Lines 345-347 "By enabling the co-extraction of high-quality microbial DNA and a comprehensive viral dsRNA signature from a single source, this protocol overcomes the fundamental challenges of spatial heterogeneity and sampling bias."

Change microbial DNA to microbial marker genes or amplicons. Also, this protocol works towards reducing spatial heterogeneity and sampling bias, but has not removed it.

Reviewer #2 (Comments for the Author):

This article puts forth a method of obtaining information on the diversity of bacteria, fungi, and viruses in one streamlined procedure in difficult to sequence soils. The authors have identified the importance and difficulty of recovering viral nucleic acids in soil in an unbiased way. The article provides some validation of the method using spiked in controls. This method will be useful to researchers - especially in well classified soils, however the strength of the method for virus recovery should be revisited.

Major concerns

The use of relative abundance and bar charts is informative, however is it hard to interpret when the total number of taxonomic units is not stated, please alter the figures so that the number of taxonomic units for each are provided, the percentage alone reduces the variation among samples that may or may not be present.

The spike control for viruses seems to be a bit overstated. The goal of the method was to capture diversity, however the virus sample has been spiked with one type of virus. I think this needs to be approached with caution, it is confirming that the method can extract RNA viruses, but not viral diversity. The bacterial spiked communities does a much better job of this. I would soften the evidence and differentiate the strengths between these two spiked tests.

Figure 4C would benefit from some restructuring, as is there is a mix of specific species, classified and unclassified, please make the lineage one specific group (such as phyla, class ?). As it is hard to tell what groups are truly dominating. The vOTU counts should also be displayed given that relative abundance is used.

As the results are right now I am not fully convinced that this method can completely characterise viruses, despite there being hits to several groups (dsRNA, RNA, and ssDNA viruses). I would expect a higher proportion of Nucleocytoviricota but depending on the filtration methods some of these would be lost as the particles are large, however it is not clear to me how the slurry was filtered - I may have missed it.

Please apply another virus identification methods or tool outside of geNomad. geNomad is very useful, however it should not be the only method of identifying viruses when the paper's purpose is to validate a method of viral diversity. Given that this was done at the contig level it may be more beneficial to run binning procedures to group contigs at least so that relative abundances are more accurate and not just many contigs from the same genome. MAGs do not need to be fully reported (annotations etc.), however bins would reduce this overrepresentation of certain viruses.

Minor Concerns

- Please discuss the mycoviruses recovered that are mentioned at line 72 in your conclusions
- Please discuss how the method has overcome humic acid contamination, line 89
- Please provide a few more details of the methods linked from protocols.io. Given that this is a methods paper it would be useful to have at least a brief overview.

We sincerely appreciate the time and effort you and the reviewers have dedicated to evaluating our work. The thoughtful and constructive feedback has been invaluable in refining and improving the clarity of our manuscript.

Please find below our point-by-point response to all reviewer comments and suggestions for the manuscript: **SS-VIME: A Single-Source Virome-Microbiome Extraction Protocol Toward Comprehensive Soil Community Analysis**. All the cited articles are listed at the end of this document.

Response to Reviewer #1

Comments for the Author. This manuscript presents and validates a new technical protocol, SS-VIME, designed to simultaneously extract microbial DNA and viral double-stranded RNA from a single soil sample. The core goal of SS-VIME is to address the fundamental challenge of spatial heterogeneity and associated sampling bias when studying virus-host dynamics in the soil's complex micro-environment. The authors successfully demonstrated proof-of-concept using spiked controls to show accurate microbial community (via amplicons) and highly specific viral signal (dsRNA fraction) recovery. Application to a vineyard soil confirms the method's ability to yield distinct microbial and virome profiles. I really appreciate the lengths the authors have gone through to make the methods easy to follow and available to others.

The methodology used in this paper does not offer the necessary empirical support for many of the strong ecological claims made in the manuscript, since many of the claims are about heterogeneity and complexity, which the authors remove when sampling with marker genes. There was no control (spike-ins were not controls here but rather proof-of-concept) or comparison to other methodology to show improvement. The paper compares their methodology to multi-omics analyses and viromics when these methods are not used or compared to. The authors should work to revise the manuscript to show a new way to extract viruses from soils along with amplicons.

Response. We thank the reviewer for their thorough assessment and constructive feedback. We appreciate the recognition of our efforts to make the protocol accessible. We agree with the reviewer's points regarding the need to temper our ecological claims and have revised the manuscript to provide more precise language regarding spatial heterogeneity (line 359-364) and viral activity. We have also added the requested methodological details and clarifications (line 178-206 and line 421-430).

Major Issues

Comment. The central claim is that the protocol "directly addresses the critical challenge of spatial heterogeneity." However, spatial heterogeneity was not directly assessed in this study, but rather, the authors demonstrated accurate recovery of spiked nucleic acids and recovery of viral

genomes. To make this claim, a comparison is needed showing reduced variance (or improved correlation between virus/host pairs) in SS-VIME data versus data generated from split/separate subsamples. I recommend revising the text to state that the protocol works towards reducing spatial heterogeneity and sampling bias but has not definitively removed or overcome it.

Response. We agree. We have revised the text throughout the manuscript (Introduction, Discussion, Conclusion) to state that the protocol "mitigates sampling bias associated with spatial heterogeneity" by enabling co-extraction from a single sample, rather than claiming it "overcomes" the heterogeneity itself. We acknowledge that heterogeneity is an inherent property of soil that cannot be removed, but our method ensures that the virome and microbiome profiles are derived from the exact same micro-environmental context.

Comment. The authors repeatedly use dsRNA as a "comprehensive proxy for the active virome." While I agree dsRNA is a good marker, the claim of capturing the "active virome" is difficult to support without functional validation. Replication takes place within a microbial cell and not all processes captured by dsRNA lead to the release of new virions or significant ecosystem impact. Increased transcriptional reads in metatranscriptomes versus controls are typically required to conclusively show metabolic activity. I recommend tempering this claim throughout the text.

Response. We appreciate the reviewer's critique regarding the definition of the 'active virome.' We agree that precision is required here. In the revised manuscript, we have explicitly defined our use of the term 'active' to refer specifically to viral replication activity. This is based on the biological reality that double-stranded RNA (dsRNA) is a molecular hallmark of genomic replication for RNA viruses and a replicative intermediate or transcriptional byproduct for DNA viruses (Aregger et al., 2012; Cottrell, Andrews, & Bass, 2024; Decker et al., 2019; Noris et al., 2023; Son, Liang, & Lipton, 2015; Weber, Wagner, Rasmussen, Hartmann, & Paludan, 2006). Its detection therefore distinguishes replicating viruses from dormant prophages or inert extracellular virions. To address the reviewer's concern about overstating ecosystem impact, we have adopted the phrase 'proxy for viral activity' in place of 'active virome' throughout much of the text (lines 26, 36, 86). Where 'active virome' is retained, we have defined it (line 376-377, 399) to ensure we are not claiming to capture the full metabolic footprint of the virus (which would require metatranscriptomics), but rather the specific signal of viral replication or transcription.

Comment. The spike-in validation is limited. No ssDNA or dsDNA viruses were spiked in, and the RNA spiked-in was only dsRNA. This does not fully mimic the stability or extraction efficiency challenges posed by encapsidated ssRNA, mRNA, or DNA viruses.

Response. We thank the reviewer for this important comment and agree that encapsidated viruses present distinct stability and extraction challenges. However, the primary objective of our method is the enrichment of viral dsRNA, which is predominantly an intracellular replication

intermediate rather than a component of mature virions. Consequently, the critical experimental bottleneck is efficient disruption of the microbial host cell, not virion lysis. To address this, our protocol employs a harsh lysis strategy (PVP–CTAB–PCI combined with bead beating), which we have previously validated for complex soil matrices containing diverse and difficult-to-lyse microorganisms (Poursalavati, Javaran, Laforest-Lapointe, & Fall, 2023). In the present study, successful recovery of DNA from the Zymo microbial community standard, including Gram-negative bacteria, Gram-positive bacteria, and yeasts [Link]) serves as a proxy for robust host cell lysis across a broad range of microbial cell types. Importantly, spiking encapsidated ssRNA or DNA viruses would not validate the dsRNA-specific capture chemistry of the cellulose column, which is the unique and defining step of our workflow. The synthetic dsRNA spike-in was therefore deliberately chosen to specifically assess the efficiency and specificity of this partitioning step. We have clarified this rationale in the revised manuscript and have explicitly discussed the scope and limitations of the method, including its focus on intracellular viral dsRNA rather than encapsidated viral genomes.

Minor Issues & Line-Specific Comments

Comment. The assessment of microbial community structure based on amplicon data has inherent limitations. While useful for quality control, marker genes cannot be used to link a virus to a host. A large part of the described bias and issues in the introduction of this paper comes from using metagenomics and metatranscriptomics to capture microbial and viral genomes to use genomic features to link them and omics is not used here at all. The amplicon work does allow for a great assessment of contamination of the virome data and is a cheap, quick means of initial characterization.

Response. We agree that amplicons cannot accurately link specific viruses to hosts. We have refined related sections and clarified that the method allows for "correlation" of community structures, not direct host-virus linking.

Comment. The Methods lack important descriptive details about the soil texture and chemistry of the vineyard soil.

Response. We appreciate the reviewer highlighting the importance of soil characterization. We have updated Section 2.2.1 (Environmental Soil Samples) to describe the key soil physicochemical properties, including the specific soil taxonomy (Blandford series, loam texture), pH, organic matter content, and C:N ratio (Lines 178-181, 183-187). Furthermore, we have compiled the complete physicochemical profile (including mineral composition and moisture levels) for both bulk and rhizosphere zones into a new Supplementary Table (Suppl. Table 1).

Comment. I am not sure what "sterilization" of the soil added to this work. If this has been validated for soil, a citation that demonstrates its effectiveness should be provided. As a

reviewer, I always argue that soils cannot be truly sterilized and attempts fundamentally alter the soil matrix.

Response. We agree that "absolute" sterilization of soil is challenging; however, our goal was to have a soil matrix free of amplifiable background nucleic acids to accurately quantify the recovery of our spike-in standards. To achieve this, we employed a rigorous tyndallization approach: three cycles of autoclaving with 24-hour incubation intervals. This method promotes the germination of heat-resistant spores during the intervals, allowing them to be killed in subsequent cycles (Otte et al., 2018; Wolf, Dao, Scott, & Lavy, 1989). Then to validate this, we performed PCR on the "sterilized" soil without spike-ins using 16S and ITS primers. As shown in Supplementary Figure 1, no amplification bands were observed, and Qubit readings were below the limit of detection. This confirms that the matrix was sufficiently devoid of interfering nucleic acids, ensuring that the signals observed in the spike-in experiments (Figure 2) originated solely from the added standards and not from the soil background. We have updated the Methods and Results sections to include these details and citations (lines 191-206).

Comment. Keywords "Soil Virome, Viromics, Soil Microbiome, dsRNA, Cellulose Column Chromatography, Unified Protocol, Single-Source Extraction" I would remove viromics, because this refers to specific methodology (e.g., virus particle separation) that was not done here. I would add Nucleic Acid Extraction

Response. We appreciate the reviewer's suggestion to improve the keywords for better discoverability. We agree that "Nucleic Acid Extraction" is a highly relevant keyword and have added it to the list. Regarding the term "Viromics" we respectfully propose retaining it. While the term was historically associated strictly with particle-enriched metagenomics, recent literature defines viromics more broadly as the sequence-based analysis of uncultivated viruses, encompassing various sampling strategies including total metagenomics and metatranscriptomics. For example, Coclet et al. define viromics analyses as "the analysis of viral genomes from metagenomes, viromes, and/or metatranscriptomes" (Coclet, Camargo Antonio, & Roux, 2024). Similarly, a recent review in Nature Reviews Genetics classifies bulk sampling and total RNA/DNA extraction workflows explicitly under "current viromics methods" (Roux & Coclet, 2026). Furthermore, our dsRNA extraction protocol is a physical enrichment method. By specifically targeting double-stranded RNA and removing host DNA and ssRNA, we are enriching the viral signal relative to the host, which aligns with the core philosophy of viromics. Therefore, we believe "Viromics" would describes the application of this method.

Comment. Lines 58-59 "This separation is profoundly problematic for soil (Geisen 2021, Liang, Radosevich et al. 2024, Hazard, Anantharaman et al. 2025)." Please add some more text to this sentence that summaries what these citations include.

Response. We thank the reviewer for this suggestion. We have expanded the text in the introduction to explicitly summarize the insights from these citations (lines 60-63). Specifically, we now highlight that the separation of extraction workflows disrupts the observation of critical

biotic interactions that occur at the micro-scale, as emphasized by Geisen and Hazard et al., and hinders the accurate assessment of viral contributions to biogeochemical cycling described by Liang et al. (Geisen, 2021; Hazard et al., 2025; Liang et al., 2024). We also added Sergaki et al. to emphasize the necessity of holistic, multi-trophic approaches for understanding soil ecosystem complexity (Sergaki, Lagunas, Lidbury, Gifford, & Schäfer, 2018).

Comment. Figure 1 legend "Downstream Processing: Each fraction is prepared for analysis. The DNA is used for amplicon PCR amplification to profile microbial communities using targeted amplicon sequencing (e.g., 16S/ITS)." For every instance where 16S is used, please always have "16S rRNA gene" or "16S rRNA amplicon".

Response. We thank the reviewer for this suggestion. We have corrected all instances to "16S rRNA gene" or "16S rRNA amplicon".

Comment. Lines 164-167 "A total of eight environmental soil samples were collected from a vineyard on mineral soil at an Agriculture and Agri-Food Canada (AAFC) experimental farm in Frelighsburg, Québec. To capture spatial micro-heterogeneity, sampling was performed at depths of 5-15 cm, targeting both the rhizosphere and the adjacent bulk soil zones." Where is the description of the soil (e.g., texture,...)? This is where the complexity and chemistry comes in and it is absent from this paper.

Comment. Lines 167-169 "Immediately following collection, all samples were transported on ice to the laboratory and stored frozen at -20°C to preserve microbial community structure and ensure nucleic acid integrity until extraction could be performed." Please describe the container the soils was stored in, if any buffers were added, and how long the soil was at -20 before nucleic acid extraction.

Response. We thank the reviewer for this suggestion. We have included details about soil, sampling and clarified the specific storage conditions to ensure reproducibility (lines 178-181, 183-187).

Comment. Lines 174-177 "These matrices were then sterilized by drying at 60°C followed by three autoclave cycles, with 24-hour incubation intervals between cycles to allow for the germination and subsequent elimination of heat-resistant endospores." If this has been validated, then please provide a citation that demonstrates effectiveness. I always argue that soils cannot be sterilized and attempts to sterilize it just alter the microbiome and virome.

Response. We have included the supporting citations and details validating the sterilization process prior to spiking the control samples (lines 194-196).

Comment. Lines 177-179 "Each sterilized matrix was then co-spiked with two standards: 75 µL of the ZymoBIOMICS Microbial Community Standard (D6300; Zymo Research, Irvine, CA, USA) to assess DNA recovery and community profile accuracy". Please briefly list the taxa in the community. It would be great to know if it is only bacterial and the phyla that is represented.

Additionally, what buffer are the spike-ins in? This was also added to the soil and would stabilize the spike-in and not represent true soil conditions.

Response. We thank the reviewer for this insightful comment regarding the composition and conditions of the spike-in controls. We have revised Section 2.2.2 to provide a detailed breakdown of the ZymoBIOMICS™ Microbial Community Standard composition, specifying that it includes eight bacterial species (spanning Firmicutes and Proteobacteria) and two fungal species (Ascomycota and Basidiomycota) (lines 199-206).

Regarding the buffer and the representation of true soil conditions: The standard is suspended in DNA/RNA Shield™. However, it is important to note that this is a whole-cell standard, not free DNA. As indicated by the manufacturer, this standard is explicitly designed to be spiked into samples to assess the efficiency of the entire workflow, starting specifically from the lysis step. The buffer serves to maintain the cellular integrity of the input prior to lysis, ensuring we are rigorously testing the protocol's ability to lyse cells (including "tough-to-lyse" Gram-positive bacteria and yeast) within the inhibitory soil matrix. Furthermore, given the small volume (75 µL) relative to the total soil mass, the buffer's effect on the overall soil chemistry is minimized, allowing the spike-in to effectively mimic the state of native microbiota during the extraction challenge.

Comment. Lines 187-190 "Following extraction, the yield and purity of the DNA and dsRNA fractions were assessed using fluorometry (Qubit; Invitrogen, Waltham, MA, USA) for accurate quantification and spectrophotometry (Nanodrop 2000; Thermo Fisher Scientific, Waltham, MA, USA) for purity ratios." Please list what version of Qubit, standards used, the amount of nucleic acid used per run, and any changes for DNA/RNA measurements. For the Nanodrop please list the purity ratios that were assessed.

Response. We appreciate the reviewer's attention to detail regarding the quantification parameters. We have updated Section 2.3.1 to explicitly list the equipment models, specific assay kits, sample input volumes, and the specific purity ratios assessed. Regarding the specific question on DNA/RNA measurement changes: We clarified that the Qubit dsDNA High Sensitivity (HS) Assay Kit was utilized for both the DNA and dsRNA fractions (lines 214-219). As no commercial Qubit assay specific for dsRNA currently exists, and standard RNA assays target single-stranded molecules, the dsDNA kit (which targets double-stranded structures) was selected to quantify the dsRNA fraction (post-DNase/RNase treatment) following a consultation with the manufacturer. This technical information is highlighted as an "Important Note" in the detailed protocol provided in Supplementary File 1 and on protocol.io.

Comment. Lines 198-200 "For the DNA fraction, amplicon libraries were generated by targeting the bacterial 16S rRNA V4-V5 region (primers 515F-Y/926R)" Please add "gene" to 16S rRNA.

Response. We thank the reviewer for this suggestion. We have added the word 'gene' to '16S rRNA' in the designated lines and have ensured this correction was applied to all similar instances throughout the revised manuscript.

Comment. Line 207 "Cutadapt". For every tool, please list the version and whether default parameters were used.

Comment. Lines 214-216 "Viral contigs were then identified using geNomad (Camargo, Roux et al. 2024) and clustered into viral operational taxonomic units (vOTUs) at 95% identity using with CD-HIT (Fu, Niu et al. 2012)." Please list what the alignment fraction is for clustering. If it is 100%, then state that.

Response. We thank the reviewer for ensuring the reproducibility of our bioinformatic workflow. We have updated Section 2.3.2 (Bioinformatic Analysis) to explicitly list the version numbers for Cutadapt and all other tools used (line 237-238). Additionally, we have specified whether default parameters were used or detailed the specific flags applied. Regarding the clustering step, we have clarified that viral contigs were clustered using CD-HIT with a 95% identity threshold and an 85% alignment coverage, following the community standards for vOTU definition established by the Minimum Information about an Uncultivated Virus Genome (MIUViG) guidelines (Roux et al., 2019). Lines 237-261.

Comment. Lines 224-225 "The method yielded high-purity DNA, with concentrations ranging from 200 to 600 ng/μl and A260/280 ratios of approximately 1.8-1.9." When extracting from soil, the concentrations need to be normalized and presented in weight of soil, typically per gram. When presenting a range please provide the standard deviation of a typical DNA extraction. Was the A230:260 not measured? Please provide if possible as this ratio also informs us on how clean the DNA is and how reliable the concentration is.

Response. We have updated Section 3.1 to present the yields as mean ± standard deviation per gram of soil (μg/g for DNA and ng/g for dsRNA), providing a specific breakdown for both bulk and rhizosphere samples. Furthermore, we have included the A260/230 ratios as requested (lines 266-271).

Comment. Lines 302-306 "In this study, we developed and validated SS-VIME, a single-source extraction protocol that successfully partitions microbial DNA and viral dsRNA from a single soil sample. This unified approach directly addresses the critical challenge of spatial heterogeneity in soil ecosystems, where 'millimeters matter' (Bi, Yu et al. 2021, Roux and Emerson 2022), and the use of separate subsamples for different targets can obscure true ecological relationships." I would revise this text. Spatial heterogeneity was not assessed here, but rather spiking in nucleic acid and recovering it. Metagenomics and a comparison to another method is needed to state this claim.

Response. We agree with the reviewer that our study validated the extraction efficiency and recovery rather than quantifying spatial heterogeneity itself. We have revised the text in the Discussion to be more precise (lines 359-364). We now state that the single-source design "mitigates" the potential sampling bias inherent to heterogeneous soils, rather than claiming to

"directly address" the heterogeneity itself. This framing more accurately reflects the methodological advantage of co-extraction without overstating the experimental scope.

Comment. Lines 316-320 "Crucially, the dsRNA fraction captured a broad virome spanning the realms Duplodnaviria (DNA viruses), Riboviria (RNA viruses), and Monodnaviria (ssDNA viruses) (Fig. 4D). The ability to recover dsRNA signatures from RNA and DNA viruses underlines the power of this method as a comprehensive proxy for the active virome, which includes viruses actively undergoing transcription and replication" While I agree this is another method to capture soil viruses, it has not been compared to other studies to show improvement. Additionally, I do not agree that the active virome is captured, even with specifying active denotes transcription and replication. Replication takes place within a microbial cell which does not always result in cell lysis and release of viruses. Additionally, it is well known that DNA and RNA can be stable in soils and thus increased transcriptional reads in metatranscriptomes vs. controls are needed to show activity.

Response. We appreciate the reviewer's critique regarding the precise definition of the 'active virome' and the context of benchmarking. We agree that precision is required to avoid overstating the ecological interpretation. In the revised manuscript, we have tempered our claims and explicitly defined our use of the term 'active' to refer specifically to viral replication and transcription activity. This distinction is grounded in the biological reality that dsRNA is a molecular hallmark of genomic replication for RNA viruses and a transcriptional byproduct for many DNA viruses (Aregger et al., 2012; Cottrell et al., 2024; Decker et al., 2019; Noris et al., 2023; Son et al., 2015; Weber et al., 2006). Consequently, we have removed the word "comprehensive" to avoid overstatement and have adopted the phrase "proxy for viral activity" in several instances. Where "active virome" is retained, it is immediately qualified by the definition of replication/transcription to distinguish it from the total virome.

Comment. Lines 323-331 "The SS-VIME protocol also offers significant versatility for broader multi-omics investigations. The optional total RNA fraction, recovered during the column wash step, provides a valuable template for metatranscriptomic analysis. This creates a powerful opportunity to link community composition (DNA) and viral replication (dsRNA) directly to host gene expression (RNA) within a single experimental framework. Moreover, the high-quality DNA fraction is not limited to bacterial and fungal profiling; it serves as a comprehensive resource for characterizing other crucial microbial guilds. With the appropriate primers, this DNA can be used to profile communities of arbuscular mycorrhizal fungi (AMF), protists, or archaea, making this protocol a flexible platform for a wide range of ecological questions." I would reduce this to a single sentence that states this method is applicable to multi-omics approaches. This work was not tested here and is a proposed use.

Response. We accept the reviewer's suggestion to condense the discussion regarding future multi-omics applications. We have reduced this section to a concise statement highlighting the

protocol's potential for RNA analysis and broader microbial profiling (e.g., AMF) without over-elaborating on untested applications (lines 384-388).

Comment. Lines 332-335 "While this protocol represents a significant technical advance, it is important to acknowledge its scope. The method is optimized to capture dsRNA as a broad and stable proxy for the active virome. Consequently, it is not designed to detect latent prophages or the inert virions of DNA and ssRNA viruses, though it will capture the genomic dsRNA from intact dsRNA virions." I would also add text about ssDNA viruses due to library methods not capturing these, viruses actively undergoing lytic infection, and not capturing microbial genomes.

Comment. Lines 335-338 "Future work should focus on testing the protocol's performance across a wider variety of challenging soil types, such as those with extremely high clay or organic matter content, and sequencing the RNA fraction to fully define its operational boundaries." High mineral soil, especially ones with lot of iron complexes create major issues, especially for RNA.

Response. We have expanded the discussion on the method's scope and limitations. We have revised the text to state that the dsRNA fraction excludes host microbial genomes and latent prophages, which is a design feature for viral signal enrichment but necessitates the parallel DNA analysis for host context. We also addressed the nuances of DNA virus detection, noting that while their dsRNA replication intermediates are captured, these viruses may be underrepresented depending on their transcriptional state. Finally, we added a specific reference to mineral-rich soils (e.g., high iron) in the future work section, acknowledging the known interference of iron complexes with RNA extraction (Lines 397-405).

Comment. Lines 345-347 "By enabling the co-extraction of high-quality microbial DNA and a comprehensive viral dsRNA signature from a single source, this protocol overcomes the fundamental challenges of spatial heterogeneity and sampling bias." Change microbial DNA to microbial marker genes or amplicons. Also, this protocol works towards reducing spatial heterogeneity and sampling bias, but has not removed it.

Response. We agree with the reviewer's suggestion to be more precise regarding the biological targets and the nature of spatial heterogeneity. We have revised the Conclusion to specify that the method co-extracts DNA suitable for "microbial marker gene profiling" rather than implying exclusive extraction of microbial DNA. Additionally, we have moderated the language regarding spatial heterogeneity, stating that the protocol "mitigates" the associated sampling bias rather than claiming to eliminate the heterogeneity itself (408-414).

Response to Reviewer #2

Comments for the Author. This article puts forth a method of obtaining information on the diversity of bacteria, fungi, and viruses in one streamlined procedure in difficult to sequence soils. The authors have identified the importance and difficulty of recovering viral nucleic acids in soil in an unbiased way. The article provides some validation of the method using spiked in controls. This method will be useful to researchers - especially in well classified soils, however the strength of the method for virus recovery should be revisited.

Response. We thank the reviewer for their positive assessment of the method's utility and their helpful suggestions.

Major Concerns

Comments. The use of relative abundance and bar charts is informative, however is it hard to interpret when the total number of taxonomic units is not stated, please alter the figures so that the number of taxonomic units for each are provided, the percentage alone reduces the variation among samples that may or may not be present.

Response. We agree with the reviewer that relative abundance can obscure differences in richness. To address this, we have modified Figure 3 and Figure 4 to display the total number of observed taxonomic units (ASVs for microbial profiles and vOTUs for viral profiles) at the top of each bar. This addition provides the necessary quantitative context to interpret the compositional variations among samples.

Comments. The spike control for viruses seems to be a bit overstated. The goal of the method was to capture diversity, however the virus sample has been spiked with one type of virus. I think this needs to be approached with caution, it is confirming that the method can extract RNA viruses, but not viral diversity. The bacterial spiked communities does a much better job of this. I would soften the evidence and differentiate the strengths between these two spiked tests.

Response. We agree with the reviewer that the single synthetic dsRNA spike-in primarily validates the specific chemical capture of the molecule rather than the breadth of viral diversity. We have revised the manuscript to clarify that the Zymo microbial community standard validates the critical lysis efficiency across diverse host cell wall types (Gram-positive, Gram-negative, and Fungi [Link]) (Lines 201-206). Since the target viral dsRNA is predominantly an intracellular intermediate, robust host lysis is the prerequisite for accessing viral diversity. Complementing this, the synthetic dsRNA spike-in validates the partitioning chemistry of the cellulose column, confirming that once released, dsRNA is efficiently captured and not lost. We have moderated the text to state that the spike-in confirms "recovery efficiency" while relying on the environmental sample data to demonstrate the method's capacity to capture "phylogenetic diversity. We have also added a statement in the Discussion acknowledging that future benchmarking with mock communities of diverse viruses would further refine the resolution of recovery rates across different viral families (Lines 380-383, 412-414).

Comments. Figure 4C would benefit from some restructuring, as there is a mix of specific species, classified and unclassified, please make the lineage one specific group (such as phyla, class ?). As it is hard to tell what groups are truly dominating. The vOTU counts should also be displayed given that relative abundance is used.

Response. We agree with the reviewer that the high proportion of unclassified "viral dark matter" typical of soil viromes can obscure the visualization of known viral diversity. To address this, we have restructured Figure 4 to focus specifically on the classified fraction of the virome. Panel 4C now displays the viral community at the Class level (rather than species), filtering out unclassified sequences and taxa with <2% relative abundance to clearly highlight the dominant known groups. Full taxonomic profiles, including the unclassified fraction, remain available in Supplementary File 5. Additionally, as requested, the total count of identified vOTUs has been added to the top of the bars to provide the necessary quantitative context.

Comments. As the results are right now I am not fully convinced that this method can completely characterise viruses, despite there being hits to several groups (dsRNA, RNA, and ssDNA viruses). I would expect a higher proportion of Nucleocytoviricota but depending on the filtration methods some of these would be lost as the particles are large, however it is not clear to me how the slurry was filtered - I may have missed it.

Response. We appreciate the reviewer's critical assessment of the method's scope. We agree that claiming to "completely characterize" the virome is an overstatement. In the revised manuscript, we have refined the text to reflect that this method specifically targets the viral fraction associated with replication and transcription, rather than the total physical virome. We have tempered our claims, explicitly defining the method as a "proxy for viral activity" as suggested by reviewer 1 (Lines 399-414). This distinction is based on the biological reality that dsRNA is a molecular hallmark of genomic replication for RNA viruses and a replicative intermediate or transcriptional byproduct for DNA viruses (Aregger et al., 2012; Cottrell et al., 2024; Decker et al., 2019; Noris et al., 2023; Son et al., 2015; Weber et al., 2006). Its detection therefore specifically distinguishes viruses actively replicating within hosts from dormant prophages or inert extracellular virions (with the exception of dsRNA viruses, where the genomic material itself can be captured).

Regarding the specific concern about *Nucleocytoviricota* (giant viruses) and filtration: We wish to clarify that this protocol differs fundamentally from standard VLP approaches in that it does not involve a size-exclusion filtration step to separate viral particles. The method relies on the direct chemical and mechanical lysis of the total soil matrix (processing host cells and viruses together). Consequently, large viral particles are not lost due to filtration. The dsRNA we capture represents the intracellular replicative intermediates or transcriptional byproducts found within the host cells. Therefore, the lower abundance of *Nucleocytoviricota* in our dataset likely reflects

the specific transcriptional activity of these DNA viruses at the time of sampling (i.e., the generation of dsRNA intermediates) rather than methodological loss.

Comments. Please apply another virus identification methods or tool outside of geNomad. geNomad is very useful, however it should not be the only method of identifying viruses when the paper's purpose is to validate a method of viral diversity. Given that this was done at the contig level it may be more beneficial to run binning procedures to group contigs at least so that relative abundances are more accurate and not just many contigs from the same genome. MAGs do not need to be fully reported (annotations etc.), however bins would reduce this overrepresentation of certain viruses.

Response. We appreciate the reviewer's suggestion to validate our viral identification methodology and ensure accurate relative abundance estimations. To address these points, we have performed two major updates to our analysis: 1) We re-processed our entire dataset using the latest, more stringent version of geNomad (v1.11.1), and 2) We performed a comprehensive cross-validation using six additional state-of-the-art tools. The results of this multi-tool comparison, including a consensus matrix and raw outputs, are provided in the new Supplementary File 3. Below, we detail our findings and the ecological justification for retaining geNomad as the primary engine for this soil dsRNA study.

Comprehensive Cross-Validation and Tool Comparison

We evaluated our data against a comprehensive suite of tools covering identification (VirSorter2, DeepVirFinder, ViraLM, TransGINmer), genome quality assessment (CheckV), and taxonomic classification (vConTACT3) (Ren, Song et al. 2020, Guo, Bolduc et al. 2021, Nayfach, Camargo et al. 2021, Peng, Shang et al. 2024, Wang, Sun et al. 2024, Bolduc, Zablocki et al. 2025). All tools were executed using default parameters. This cross-comparison revealed distinct performance profiles (see Supplementary File 3):

Sensitivity vs. Specificity Trade-off. Pure machine-learning tools (e.g., TransGINmer [11,702 hits], DeepVirFinder [6,368 hits]) identified high volumes of hits; however, previous benchmarks indicate these k-mer-based approaches suffer from higher false-positive rates in complex biomes like soil, often confusing eukaryotic repetitive elements with viral signatures (Wu et al., 2024). Conversely, marker-based tools (VirSorter2 [283 hits], CheckV [522 hits]) were overly conservative, missing novel RNA viruses lacking standard homologs. As noted in recent literature, this creates a "reference trap" where novel soil viruses are rejected simply because they lack homology to known lineages in RefSeq (Camargo et al., 2024; Feeser, Longley, Gallegos-Graves La, Albright, & Shakya, 2025).

geNomad Selection. We retained geNomad (2,798 hits) as our primary engine because of its hybrid architecture: it combines the IGLOO neural network (which detects viral nucleotide "texture" independent of gene content) with a massive marker database of 227,897 protein profiles (Camargo et al., 2024). This dual approach allows it to detect divergent RNA viruses that lack standard homologs required by VirSorter2, while using the marker branch to filter out the

noise common in DeepVirFinder. The reliability of geNomad for large-scale discovery in diverse ecosystems has been further validated by its use in constructing the massive VIRE database (Nishijima, Fullam, Schmidt, Kuhn, & Bork, 2025). To ensure robustness, we upgraded to geNomad v1.11.1, increasing the minimum score threshold from 0.5 to 0.7 and enforcing stricter post-classification filters (e.g., requiring hallmark genes for sequences <2,500 bp). This significantly reduces the risk of false positives compared to older versions while maintaining sensitivity.

Performance on Short Sequences. dsRNA extraction often yields short and fragmented contigs, and benchmarking shows that legacy tool performance degrades precipitously on contigs <5kb. However, geNomad maintains high sensitivity (~90%) on short fragments by leveraging attention mechanisms to weight nucleotide composition over missing marker genes (Camargo et al., 2024; Wu et al., 2024).

The "Consensus" Trap. We specifically avoided filtering our dataset to only include contigs identified by multiple tools (e.g., a "majority vote"). Recent benchmarking explicitly cautions against this, noting that because tools rely on different signals (homology vs. composition), intersecting them often discards valid, novel viruses (the "lowest common denominator" effect) rather than improving accuracy (Hegarty et al., 2024).

Addressing Binning and Abundance Accuracy

We agree with the reviewer that ensuring accurate relative abundance and mitigating the overrepresentation of viral genomes is critical. We carefully considered the application of multi-contig genome binning (e.g., using tools like vRhyme or PHAMB); however, based on recent literature and the specific nature of our dataset, we determined that a contig-level analysis with rigorous de-replication was the most scientifically robust approach for this study. This is based on two primary technical constraints inherent to environmental RNA viromes:

Challenges with Segmented Genomes. Unlike DNA bacteriophages, where binning is well-established, RNA viruses frequently possess segmented genomes. Recent benchmarking by Tang et al. demonstrates that standard binning tools often fail to reconstruct segmented viruses because k-mer distributions can vary significantly between different segments of the same virus (Tang et al., 2024). Consequently, binning algorithms risk classifying valid viral segments as contamination or "misbinned" contigs, leading to artificial data loss. As noted in Debat and Bejerman, the discovery of novel segmented families is expanding, and forcing these distinct segments into single bins without reference homology risks obscuring this diversity (Debat & Bejerman, 2025).

Sequence Length Limitations. As noted, our dataset contains a significant proportion of valid viral fragments shorter than 2kb. Current state-of-the-art viral binning tools have distinct length limitations. For instance, Kieft et al. explicitly state that their tool, vRhyme, uses a minimum input size of 2kb for optimal results (Kieft, Adams, Salamzade, Kalan, & Anantharaman, 2022), and Johansen et al. similarly focus on medium-to-high quality genomes which often excludes

smaller fragments (Johansen et al., 2022). Applying these thresholds to our data would inadvertently exclude a large portion of the active RNA virome, particularly fragmented sequences typical of dsRNA metagenomic assemblies.

Mitigating Overrepresentation via Clustering. To address the reviewer's valid concern regarding the potential overrepresentation of certain viruses, we employed a rigorous clustering strategy as an alternative to binning. As noted by Tadmor and Phillips, assembly-based viral detection is inherently prone to redundancy due to fragmented assemblies and high mutation rates, which necessitates data compression (Tadmor & Phillips, 2022). To correct for this, we clustered all identified viral contigs into viral Operational Taxonomic Units (vOTUs) using CD-HIT. We applied clustering thresholds of 95% sequence identity and 85% alignment coverage, adhering to the community standards for vOTU definition established by the Minimum Information about an Uncultivated Virus Genome (MIUViG) guidelines (Roux et al., 2019). This methodology aligns with standards established in recent virome benchmarks and studies which utilized CD-HIT for vOTU dereplication (Guo et al., 2025; Li et al., 2025; Sato, 2025; Sato, Kumagai, Hirooka, & Yoshida, 2025; Sun et al., 2025; Zhang et al., 2024; Zheng, Gao, Wu, & Ruan, 2024). This de-replication step effectively collapses redundant contigs, overlapping fragments, and strain-level variants into representative vOTUs. While distinct from genome binning, this approach directly addresses the goal of mitigating diversity inflation caused by redundancy, ensuring that our abundance profiles reflect the viral community structure without the risks of false-exclusion associated with binning short or segmented RNA viral fragments.

Minor Concerns

Comments. Please discuss the mycoviruses recovered that are mentioned at line 72 in your conclusions

Response. We appreciate the reviewer's suggestion to highlight these specific biological findings. We have revised the Conclusions section to discuss the recovery of mycoviruses, noting the detection of families such as *Polymycoviridae* and *Betaormycoviridae* in our environmental data. This addition validates the protocol's specific utility for studying fungal-viral interactions in complex soil ecosystems (Line 424-427).

Comments. Please discuss how the method has overcome humic acid contamination, line 89

Response. We agree that specifying the mechanism for humic acid removal adds valuable context. We have updated the Introduction to briefly describe the chemical and physical parameters used to mitigate contamination. Specifically, we highlighted the synergistic use of Polyvinylpyrrolidone (PVP) to bind polyphenols, optimized phosphate buffering, low-temperature processing to limit humic solubility, and selective PEG precipitation, all of which were established in our previous published methodology (Poursalavati, Javaran et al. 2023) (Lines 91-96).

Comments. Please provide a few more details of the methods linked from protocols.io. Given that this is a methods paper it would be useful to have at least a brief overview.

Response. We thank the reviewer for their suggestion. We agree that providing a more detailed overview of the methodology directly within the manuscript is beneficial for readers. We have expanded the Methods section to include key details of the lysis buffer components, the specific enzymatic treatments, and the sequential recovery process for both DNA and dsRNA fractions, while still referencing the full protocol for comprehensive step-by-step instructions (Lines 163-173).

References

- Aregger, M., Borah, B. K., Seguin, J., Rajeswaran, R., Gubaeva, E. G., Zvereva, A. S., et al. (2012). Primary and Secondary siRNAs in Geminivirus-induced Gene Silencing. *PLOS Pathogens*, *8*(9), e1002941.
- Camargo, A. P., Roux, S., Schulz, F., Babinski, M., Xu, Y., Hu, B., et al. (2024). Identification of mobile genetic elements with geNomad. *Nature Biotechnology*, *42*(8), 1303-1312.
- Coclet, C., Camargo Antonio, P., & Roux, S. (2024). MVP: a modular viromics pipeline to identify, filter, cluster, annotate, and bin viruses from metagenomes. *mSystems*, *9*(10), e00888-00824.
- Cottrell, K. A., Andrews, R. J., & Bass, B. L. (2024). The competitive landscape of the dsRNA world. *Molecular Cell*, *84*(1), 107-119.
- Debat, H., & Bejerman, N. (2025). An Update on RNA Virus Discovery: Current Challenges and Future Perspectives. *Viruses*, *17*(7), 983. doi:10.3390/v17070983
- Decker, C. J., Steiner, H. R., Hoon-Hanks, L. L., Morrison, J. H., Haist, K. C., Stabell, A. C., et al. (2019). dsRNA-Seq: Identification of Viral Infection by Purifying and Sequencing dsRNA. *Viruses*, *11*(10). doi:10.3390/v11100943
- Feeser, K., Longley, R., Gallegos-Graves La, V., Albright, M., & Shakya, M. (2025). Recovering new viruses from New Mexico soils. *Microbiology Resource Announcements*, *14*(11), e00908-00925.
- Geisen, S. (2021). The Future of (Soil) Microbiome Studies: Current Limitations, Integration, and Perspectives. *mSystems*, *6*(4), e0061321.
- Guo, X., Liang, Y., Gao, C., Yu, H., Wang, M., Shao, H., et al. (2025). Diversity and Ecological Potentials of Marine Viruses Inhabiting Continental Shelf Seas. *Advanced Science*, *n/a*(n/a), e11707.

- Hazard, C., Anantharaman, K., Hillary, L. S., Neri, U., Roux, S., Trubl, G., et al. (2025). Beneath the surface: Unsolved questions in soil virus ecology. *Soil Biology and Biochemistry*, 205, 109780.
- Hegarty, B., Riddell V, J., Bastien, E., Langenfeld, K., Lindback, M., Saini, J. S., et al. (2024). Benchmarking informatics approaches for virus discovery: caution is needed when combining in silico identification methods. *Msystems*, 9(3), e01105-01123.
- Johansen, J., Plichta, D. R., Nissen, J. N., Jespersen, M. L., Shah, S. A., Deng, L., et al. (2022). Genome binning of viral entities from bulk metagenomics data. *Nature Communications*, 13(1), 965.
- Kieft, K., Adams, A., Salamzade, R., Kalan, L., & Anantharaman, K. (2022). vRhyme enables binning of viral genomes from metagenomes. *Nucleic Acids Research*, 50(14), e83-e83.
- Li, Z., Liu, B., Cao, B., Cun, S., Liu, R., & Liu, X. (2025). The potential role of viruses in antibiotic resistance gene dissemination in activated sludge viromes. *Journal of Hazardous Materials*, 486, 137046.
- Liang, X., Radosevich, M., DeBruyn, J. M., Wilhelm, S. W., McDearis, R., & Zhuang, J. (2024). Incorporating viruses into soil ecology: a new dimension to understand biogeochemical cycling. *Critical reviews in environmental science and technology*, 54(2), 117-137.
- Nishijima, S., Fullam, A., Schmidt, T. S. B., Kuhn, M., & Bork, P. (2025). VIRE: a metagenome-derived, planetary-scale virome resource with environmental context. *Nucleic Acids Research*, gkaf1225.
- Noris, E., Pegoraro, M., Palzhoff, S., Urrejola, C., Wochner, N., Kober, S., et al. (2023). Differential Effects of RNA-Dependent RNA Polymerase 6 (RDR6) Silencing on New and Old World Begomoviruses in *Nicotiana benthamiana*. *Viruses*, 15(4).
- Otte, J. M., Blackwell, N., Soos, V., Rughöft, S., Maisch, M., Kappler, A., et al. (2018). Sterilization impacts on marine sediment---Are we able to inactivate microorganisms in environmental samples? *FEMS Microbiology Ecology*, 94(12), fyy189.
- Poursalavati, A., Javaran, V. J., Laforest-Lapointe, I., & Fall, M. (2023). Soil metatranscriptomics: An improved RNA extraction method toward functional analysis using nanopore direct RNA sequencing. *Phytobiomes Journal*.
- Roux, S., Adriaenssens, E. M., Dutilh, B. E., Koonin, E. V., Kropinski, A. M., Krupovic, M., et al. (2019). Minimum information about an uncultivated virus genome (MIUViG). *Nature biotechnology*, 37(1), 29-37.
- Roux, S., & Coclet, C. (2026). Viromics approaches for the study of viral diversity and ecology in microbiomes. *Nature Reviews Genetics*, 27(1), 32-46.
- Sato, Y. (2025). Rumen DNA virome in beef cattle reveals an unexplored diverse community with potential links to carcass traits. *ISME Communications*, 5(1), ycaf021.

- Sato, Y., Kumagai, H., Hirooka, H., & Yoshida, T. (2025). Differences in prokaryotic and viral community between rumen and feces. *Scientific Reports*, *15*(1), 43232.
- Son, K. N., Liang, Z., & Lipton, H. L. (2015). Double-Stranded RNA Is Detected by Immunofluorescence Analysis in RNA and DNA Virus Infections, Including Those by Negative-Stranded RNA Viruses. *J Virol*, *89*(18), 9383-9392.
- Sun, R., Xu, W., Xu, Y., Xu, Z., Tan, Y., Li, J., et al. (2025). Environmental gradients shape viral-host dynamics in the Pearl River estuary. *ISME Communications*, *5*(1), ycaf164.
- Tadmor, A. D., & Phillips, R. (2022). MCRL: using a reference library to compress a metagenome into a non-redundant list of sequences, considering viruses as a case study. *Bioinformatics*, *38*(3), 631-647.
- Tang, X., Shang, J., Chen, G., Chan, K. H. K., Shi, M., & Sun, Y. (2024). SegVir: Reconstruction of Complete Segmented RNA Viral Genomes from Metatranscriptomes. *Molecular Biology and Evolution*, *41*(8), msae171.
- Weber, F., Wagner, V., Rasmussen, S. B., Hartmann, R., & Paludan, S. R. (2006). Double-stranded RNA is produced by positive-strand RNA viruses and DNA viruses but not in detectable amounts by negative-strand RNA viruses. *J Virol*, *80*(10), 5059-5064.
- Wolf, D. C., Dao, T. H., Scott, H. D., & Lavy, T. L. (1989). *Influence of sterilization methods on selected soil microbiological, physical, and chemical properties* (No. 0047-2425): Wiley Online Library.
- Wu, L.-Y., Wijesekara, Y., Piedade, G. J., Pappas, N., Brussaard, C. P. D., & Dutilh, B. E. (2024). Benchmarking bioinformatic virus identification tools using real-world metagenomic data across biomes. *Genome Biology*, *25*(1), 97.
- Zhang, Q., Xiong, Y., Zhang, J., Liu, B., Chen, T., Liu, S., et al. (2024). Eutrophication impacts the distribution and functional traits of viral communities in lakes. *Science of The Total Environment*, *946*, 174339.
- Zheng, Y., Gao, Z., Wu, S., & Ruan, A. (2024). Community Structure, Drivers, and Potential Functions of Different Lifestyle Viruses in Chaohu Lake. *Viruses*, *16*(4), 590. doi:10.3390/v16040590

Re: Spectrum03323-25R1 (**SS-VIME: A Single-Source Virome-Microbiome Extraction Protocol Toward Comprehensive Soil Community Analysis**)

Dear Dr. Mamadou Lamine Fall:

Your manuscript has been accepted, and I am forwarding it to the ASM production staff for publication. Your paper will first be checked to make sure all elements meet the technical requirements. ASM staff will contact you if anything needs to be revised before copyediting and production can begin. Otherwise, you will be notified when your proofs are ready to be viewed.

Sincerely,
Blair Steven
Editor
Microbiology Spectrum